# Low-Dose Acrolein, an Endogenous and Exogenous Toxic Molecule, Inhibits Glucose Transport via an Inhibition of Akt-Regulated GLUT4 Signaling in Skeletal Muscle Cells

**DOI:** 10.3390/ijms22137228

**Published:** 2021-07-05

**Authors:** Ching-Chia Wang, Huang-Jen Chen, Ding-Cheng Chan, Chen-Yuan Chiu, Shing-Hwa Liu, Kuo-Cheng Lan

**Affiliations:** 1Department of Pediatrics, College of Medicine, National Taiwan University & Hospital, Taipei 100, Taiwan; ccwangy1@ntu.edu.tw; 2Institute of Toxicology, College of Medicine, National Taiwan University, Taipei 100, Taiwan; r02447001@ntu.edu.tw; 3Department of Geriatrics and Gerontology, College of Medicine, National Taiwan University, Taipei 100, Taiwan; dingchengchan@ntu.edu.tw; 4Center of Consultation, Center for Drug Evaluation, Taipei 115, Taiwan; kidchiou@gmail.com; 5Department of Medical Research, China Medical University Hospital, China Medical University, Taichung 404, Taiwan; 6Department of Emergency Medicine, Tri-Service General Hospital, National Defense Medical Center, Taipei 114, Taiwan

**Keywords:** acrolein, skeletal muscle, glucose metabolism, Akt, glucose transporter

## Abstract

Urinary acrolein adduct levels have been reported to be increased in both habitual smokers and type-2 diabetic patients. The impairment of glucose transport in skeletal muscles is a major factor responsible for glucose uptake reduction in type-2 diabetic patients. The effect of acrolein on glucose metabolism in skeletal muscle remains unclear. Here, we investigated whether acrolein affects muscular glucose metabolism in vitro and glucose tolerance in vivo. Exposure of mice to acrolein (2.5 and 5 mg/kg/day) for 4 weeks substantially increased fasting blood glucose and impaired glucose tolerance. The glucose transporter-4 (GLUT4) protein expression was significantly decreased in soleus muscles of acrolein-treated mice. The glucose uptake was significantly decreased in differentiated C2C12 myotubes treated with a non-cytotoxic dose of acrolein (1 μM) for 24 and 72 h. Acrolein (0.5–2 μM) also significantly decreased the GLUT4 expression in myotubes. Acrolein suppressed the phosphorylation of glucose metabolic signals IRS1, Akt, mTOR, p70S6K, and GSK3α/β. Over-expression of constitutive activation of Akt reversed the inhibitory effects of acrolein on GLUT4 protein expression and glucose uptake in myotubes. These results suggest that acrolein at doses relevant to human exposure dysregulates glucose metabolism in skeletal muscle cells and impairs glucose tolerance in mice.

## 1. Introduction

Insulin increases glucose uptake into adipocytes and skeletal muscle cells to control blood glucose levels in the circulation via a cascade of signal transduction activation, which triggers glucose transporter-4 (GLUT4) from the cytosol to the cell membrane [1,2]. Skeletal muscle comprises 40% to 45% of adult human body weight and has remarkable plasticity, which can adapt to physiological and environmental influences by changing the characteristics of contraction and metabolism [3,4]. Skeletal muscle undergoes a regenerative process that involves tissue, cellular, and molecular signals when responding to injury [5]. Skeletal muscle is an important tissue that maintains glucose homeostasis through glucose transporter-4 (GLUT4) by translocation to plasma membrane and facilitating glucose uptake [6,7]. GLUT4 is known as the major insulin-mediated glucose transporter in muscle and adipose tissues. The expression of GLUT4 is dependent on the insulin receptor-mediated phosphatidylinositol 3-kinase (PI3K)/Akt pathway [8]. The impairment of glucose transport in skeletal muscle is a major factor responsible for reduced glucose uptake in type 2 diabetic patients [9].

Acrolein is known to be a small molecule and an extremely electrophilic aldehyde from endogenous and exogenous sources [10]. The exogenous sources of acrolein include food, the environment, and tobacco [10]. High levels of cooking-related acrolein have been detected in grocery stores [11]. One endogenous source of acrolein is the metabolites of cyclophosphamide, a chemotherapy drug [10]. Acrolein protein adducts have been demonstrated to be associated with the pathogenesis of diabetic complications, such as nephropathy and retinopathy [12,13]. Acrolein at a concentration of 10 μM has been found to induce the attenuation in endothelial cell migration and insulin signaling in cultured human umbilical vein endothelial cells [14]. Feroe et al. have shown a positive association of urinary acrolein metabolites N-acetyl-S-(3-hydroxypropyl)-l-cysteine (3-HPMA) and N-acetyl-S-(carboxyethyl)-l-cysteine (CEMA) and their molar sum (Σ acrolein) with diabetes and insulin resistance using data from a National Health and Nutrition Examination Survey (NHANES) (2005–2006) [15]. Recently, acrolein at concentrations of 0.125–1 μM has been shown to significantly suppress myogenic differentiation in C2C12 myoblasts via inhibition of Akt signaling [16]. They also found that acrolein (2.5 and 5 mg/kg) can induce skeletal muscle wasting and retard muscle regeneration in a mouse model. However, the detailed effects and molecular mechanisms of acrolein on glucose metabolism or glucose transport signaling in the skeletal muscle is still unclear and needs to be further clarified.

Workers in bars and taverns have been estimated to be exposed to acrolein from cigarette smoke at concentrations of 15 to 1830 μg/day [17]. The concentration of acrolein in cigarette smoke has been measured to be up to 90 ppm [18]. Higher mean concentrations of free acrolein and acrolein-protein adducts in plasma measuring 1.02–1.42 μM and 138–170 μM (moderate to severe), respectively, have been found in patients with chronic renal failure as compared to about 0.5 μM and 30 μM, respectively, in normal subjects [19]. The peak blood acrolein concentrations in patients administered with cyclophosphamide (60 mg/kg/day) by 1-h infusion for 2 days has been shown to be 6.2–10.2 μM [17]. Therefore, this study investigated whether acrolein at doses relevant to human exposure affects muscular glucose metabolism in vitro (0.5–2 μM acrolein) and glucose tolerance in vivo (2.5 and 5 mg/kg acrolein).

## 2. Results

### 2.1. The In Vivo Effects of Acrolein on Blood Glucose and Muscle GLUT4 Expression in Mice

The effect of acrolein on fasting blood glucose in mice was first tested. As shown in Figure 1A, the fasting blood glucose levels were significantly elevated by acrolein (2.5 and 5 mg/kg) exposure for 4 weeks in mice. The blood glucose curve and area under the glucose curve (AUC) during the oral glucose tolerance test (OGTT) also showed that the glucose tolerance was impaired in acrolein-treated mice (Figure 1B). Acrolein (2.5 and 5 mg/kg) exposure also significantly increased the insulin levels in blood samples and HOMA-IR (Figure 1C). Skeletal muscle has an essential role in whole-body glucose homeostasis in which GLUT4 is an indispensable factor [20]. The effect of acrolein on GLUT4 expression in the mouse skeletal muscle was next investigated. As shown in Figure 1D, the protein expression of GLUT4 was significantly and markedly decreased in the soleus muscle of acrolein-treated mice.

### 2.2. The In Vitro Effects of Acrolein on Glucose Metabolic Signaling in C2C12 Myotubes

Differentiated C2C12 myotube has been suggested to be a convenient in vitro model for glucose metabolism analysis in skeletal muscle [21]. Here, the effects of acrolein on glucose metabolism in C2C12 myotubes were tested. After differentiation for 4 days, C2C12 myotubes were exposed to acrolein (1 μM) for 24 and 72 h. The glucose uptake determined by 2-NBDG assay was significantly decreased in myotubes treated with acrolein for 24 and 72 h (Figure 2A). Insulin (10 nM) as a positive control significantly increased the glucose uptake in myotubes (Figure 2A). The effect of acrolein on the protein expressions of GLUT4 and related signaling molecules in myotubes was further investigated. As shown in Figure 1C and Figure 2B, treatment with acrolein (0.5–1 μM) for 24 and 72 h significantly decreased the GLUT4 protein expression in myotubes. Moreover, we also tested the effects of acrolein on glucose metabolism-related signaling molecules in myotubes. As shown in Figure 3A, the levels of phosphorylation for insulin receptor (IR) substrate 1 (IRS1), Akt, mTOR, p70S6K, and glycogen synthase kinase 3 (GSK3)α/β in myotubes were suppressed by acrolein (1 μM) treatment for 24 h. The phosphorylation of p85/PI3K in myotubes was also reduced by acrolein (1 μM) treatment for 24 h (Figure 3B). Acrolein (1 μM) treatment significantly increased the glycogen content in myotubes (Figure 3B).

We next examined the effects of acrolein in insulin signaling in C2C12 myotubes. Myotubes were cultured for 24 h with acrolein (1 μM), and then stimulated with insulin (10 nM) for 10 min. As shown in Figure 4, the levels of phosphorylation for IRS1, Akt, and GSK3α/β, but not insulin receptor (IR), were significantly reduced by acrolein (1 μM) treatment in insulin-stimulated myotubes.

Activation of Akt promotes the mobilization of GLUT4 vesicles to the plasma membrane in the skeletal muscle cells [22]. Here, the role of Akt signaling in the inhibitory effect of acrolein on GLUT4 expression was investigated. Transfection of the constitutively active form of Akt significantly reversed the inhibitory effects of acrolein (1 μM) on GLUT4 and phosphorylated GSK3α/β expression in myotubes (Figure 5). As shown in Figure 6A, transfection of the constitutively active form of Akt significantly reversed the inhibitory effect of acrolein (1 μM) on GLUT4 expressions in both cytosol fraction and plasma membrane fraction of myotubes. Over-expression of the constitutively active form of Akt can also prevent the inhibitory effect of acrolein on glucose uptake in myotubes (Figure 6B).

The effects of acrolein exposure on the Akt signaling in soleus muscle of mice were next tested. As shown in Figure 7, the immunohistochemistry staining showed that the soleus muscle isolated from acrolein-treated mice had decreased phosphorylated Akt protein expression.

## 3. Discussion

It has been estimated that about 80% of glucose uptake occurs in the skeletal muscle under euglycemic hyper-insulinemic conditions [23]. Glucose utilization plays an essential role in myogenesis and GLUT4 expresses predominantly in the skeletal muscle to respond to glucose uptake [24]. In the present study, the impairment of acrolein by oral gavage (2.5 and 5 mg/kg) on the blood glucose level and OGTT was observed in vivo. The low-dose acrolein (1 μM) also impaired glucose homeostasis by inhibition of protein expressions of glucose metabolism-related signals on skeletal myotubes in vitro. These results suggest that acrolein exposure at doses relevant to human exposure is capable of impairing glucose transport in the skeletal muscle cells and inducing hyperglycemia in mice.

The murine C2C12 skeletal muscle cell line is one of the most widely used cell lines in skeletal muscle tissue engineering [25]. The differentiated C2C12 myotube is a convenient in vitro model for analysis of glucose uptake [21]. GLUT4 is a predominantly expressed isoform in the skeletal muscle and responds to glucose transport through translocation from the intracellular site to the surface membrane [26]. Insulin-triggered GLUT4 translocation to the plasma membrane requires IRS1-associated PI3K activation, which subsequently activates Akt signaling [27]. Tremblay et al. have found that insulin-stimulated glucose transport and IRS1-associated PI3K/Akt activity, but not insulin-induced tyrosine phosphorylation of the insulin receptor and IRS proteins, are significantly impaired in the skeletal muscles of high fat diet-fed rats [28]. Akt-mediated mTOR activation activates its downstream proteins’ phosphorylation, such as p70S6K, to trigger protein synthesis in the skeletal muscle [29]. O’Toole et al. (2014) have also demonstrated that acrolein reduces Akt phosphorylation in response to insulin and decreases insulin sensitivity in endothelial cells [14]. Our previous study also demonstrated that acrolein exposure in mice can decrease the phosphorylated Akt protein expression in the skeletal muscle [16]. In the present study, we found that GLUT4 protein expression was markedly decreased in the skeletal muscle of acrolein-treated mice. Low-concentration acrolein (1 μM) also significantly reduced the protein expression of GLUT4 at 24 h treatment and inhibited the recruitment and biosynthesis of GLUT4 within 72 h in C2C12 myotubes. The phosphorylation of IRS1, Akt, mTOR, and p70S6K was also suppressed by acrolein (1 μM) in C2C12 myotubes. Moreover, transfection of the constitutively active form of Akt significantly ameliorated the inhibitory effect of acrolein on GLUT4 protein expression or translocation to the membrane. Consistent with the previous findings, we also found that the phosphorylation of Akt in soleus muscle was reduced in acrolein-treated mice. These results suggest that acrolein retards the translocation and biosynthesis of GLUT4 via inhibition in the Akt-dependent pathway.

Phosphorylation of IRS1 Ser307 has been shown to inhibit the interaction between IRS1 and insulin receptor, leading to inhibit insulin action in a 32D myeloid progenitor cell model [30]. Hançer et al. have found that three of the IRS1 Ser(P) residues 302, 307, and 612 are negatively correlated with tyrosine phosphorylation of IRS1 in a Chinese hamster ovary-insulin receptor/IRS1 over-expression cell model, showing an insulin-desensitizing effect [31]. However, contrary to the results from experiments of cell models, Copps et al. found that IRS1 Ser307 played a positive regulatory role in maintaining insulin signaling in a separate knock-in mouse model, resulting in moderating the severity of insulin resistance [32]. Therefore, more evidence is needed to clarify this controversial issue. On the other hand, Hançer et al. have shown that the inhibition of the PI3K-Akt-mTOR pathway by pharmacological inhibitors can consistently inhibit IRS1 Ser(P) 265, 302, 307, 522, and 632 [31]. The present study focused on the role of PI3K/Akt in acrolein-inhibited GLUT4-related glucose uptake. Our results revealed that acrolein inhibited the activation of PI3K-Akt-mTOR signaling pathway in myotubes. Acrolein also inhibited the levels of phosphorylation for Akt, GSK3α/β, and IRS1, but not IR, in insulin-stimulated myotubes. These findings indicated that acrolein exposure could interfere with the insulin signaling in myotubes. Our results emphasized the important role of PI3K/Akt signaling in acrolein-impaired glucose uptake in skeletal muscles. However, the detailed effects and mechanisms of acrolein on the inhibition of IRS1 Ser(P) 307 and insulin receptor function need further investigation in the future.

GSK3 signaling is known to be involved in several cellular functions: protein synthesis, glycogen synthesis, gene transcription, and cell differentiation [33]. The activity of GSK3 can be inhibited by serine phosphorylation in response to insulin mediated by Akt [34]. The evidence showed that GSK3 overactivity plays a role in the development of insulin resistance of glucose transport and glycogenesis [33]. GSK3 overactivity in obesity impairs IRS-1-dependent signaling and reduces GLUT4 translocation and glucose transport activity in the skeletal muscle [33]. GSK3α and GSK3β are two isoforms of GSK3. The inhibition of both forms of GSK3 has been demonstrated to be necessary for insulin re-sensitization in muscles and livers of type 2 diabetes [35]. Moreover, glycogen plays an essential role in maintaining a physiological blood glucose concentration and is regulated by the balance between the glycogen synthase and glycogen phosphorylase activities in the skeletal muscle [36]. The impairment of glycogen synthase results in less glycogen accumulation in the skeletal muscle. The present study found that low-concentration acrolein significantly inhibited the phosphorylation of GSK3α/β in C2C12 myotubes, which could be effectively reversed by transfection of the constitutively active form of Akt. These results suggest that the inhibition of Akt-regulated GSK3α/β phosphorylation by acrolein activates GSK3 activity, which may further retard glucose transport and glycogen synthesis in myotubes. However, we observed a considerable increase in glycogen content in acrolein-treated myotubes. Therefore, the increased glycogen content by acrolein in myotubes cannot be explained by an increase in the activity of the classical insulin-signaling pathway (PI3K-Akt-GSK3). These results indicate that other pathways may be involved in the modulation of glycogen synthase in acrolein-treated myotubes. The glycogen content in muscles has been found to be effectively increased in GLUT4 null mice [37]. Kim et al. have also found that the muscular glycogen synthase activity and glycogen content in muscle-specific GLUT4 knockout (KO) mice are markedly elevated despite a 75% reduction in glucose uptake [38]. They found that PI3K and Akt activities were decreased and GSK3β activity was increased, predicting that glycogen synthesis would be inhibited in GLUT4-KO muscles; however, the glycogen content was actually increased [38]. They further demonstrated that the increased hexokinase II, glucose-6-phosphate, muscle-specific regulatory subunit (RGL), protein targeting to glycogen (PTG) levels and enhanced glycogen-targeting subunits of protein phosphatase 1 (PP1) activity might contribute to the increased glycogen synthase activity and glycogen content [38]. Acrolein has been shown to increase the liver glycogen content of fasted rats, which may be associated with increased adrenal glucocorticoids, leading to the stimulation of gluconeogenesis and glycogenesis [39]. Our findings revealed that acrolein exposure significantly decreased GLUT4 protein expression and glucose uptake, but markedly increased glycogen contents in myotubes, suggesting that low-concentration acrolein exposure may retard glucose metabolism and evoke the compensative glycogen synthesis in myotubes, which may be independent of the PI3K-Akt-GSK3 pathway. Nevertheless, the real mechanism for this issue needs to be clarified in the future.

## 4. Materials and Methods

### 4.1. Animals

Male ICR mice (25–30 g) were obtained from the Animal Center of the College of Medicine, National Taiwan University (Taipei, Taiwan). The ethical review committee of the College of Medicine, National Taiwan University approved this animal study. The experiments were conducted in accordance with the Guide for the Care and Use of Laboratory Animals [40]. Mice were separately housed in cages maintained at a temperature of 22 ± 2 °C and 12-h light-dark cycles with free access to diet and water. Mice were humanely treated and with regard for the alleviation of suffering. Mice were randomly divided into 3 groups (8 mice/group) and exposed to acrolein in distilled water by oral gavage (0, 2.5, or 5 mg/kg) daily for 4 weeks.

### 4.2. Blood Glucose Analysis and Oral Glucose Tolerance Test

Blood samples were collected via the tail vein in fasting mice. The levels of blood glucose were determined by an Antsense-III glucose analyzer (Horiba Industry, Kyoto, Japan). A glucose tolerance test was performed on fasting mice. Mice were given a glucose challenge (2 g/kg) by oral gavage. Blood samples were collected before and 15, 30, 60, 90, and 120 min after delivery of the glucose intake. OGTT was calculated and demonstrated as area under the curve (AUC). The plasma insulin detection was performed by an immunoassay kit (Mercodia, Uppsala, Sweden). HOMA-IR = (FPI × FPG)/22.5; FPI: fasting plasma insulin concentration (µU/L); FPG: fasting plasma glucose (mmol/L).

### 4.3. Myoblasts and Myotube Formation

The murine skeletal muscle myoblast cell line C2C12 was purchased from American Type Culture Collection (Manassas, VA, USA). C2C12 myoblasts were maintained in growth medium (GM) with Dulbecco’s modified Eagle’s medium (DMEM) supplemented with 10% heat-inactivated fetal bovine serum (FBS) and antibiotics (100 IU/mL penicillin, 100 μg/mL streptomycin and 0.25 μg/mL amphotericin B) at 37 °C and 5% CO_2_ in a humidified atmosphere.

The myogenic differentiation was determined as previously described in [16]. C2C12 cells were cultured in differentiation medium (DM) (MCDB201 and Ham’s F12 medium; Invitrogen, Carlsbad, CA, USA) along with 2% heat-inactivated horse serum to induce differentiation and myotube formation.

### 4.4. Glucose Transport Assay

The glucose transport assay was performed on differentiated C2C12 myotubes. The glucose uptake activity was detected using 2-[N-(7-nitrobenz-2-oxa-1,3-diazol-4-yl)-amino]-2-deoxy-D-glucose (2-NBDG; Life Technologies, Grand Island, NY, USA), a fluorescent D-glucose analog, as previously described [41]. Briefly, myotubes were pre-cultured in Krebs Henseleit buffer (pH 7.4 with 2% BSA) containing 0.05% glucose at 37℃ for 30 min. Myotubes were then treated with 2-NBDG (200 μM) for 30 min in glucose-free Krebs Henseleit buffer with 2% BSA. The fluorescence intensities of 2-NBDG uptake were measured by a microplate fluorometer (Beckman Coulter, Brea, CA, USA) with λex = 485 nm and λem = 535 nm.

### 4.5. Protein Extraction and Western Blot and Immunohistochemical Analysis

Protein extraction and Western blot analysis were determined as previously described [16]. Myotubes were harvested by scraping with a radioimmunoprecipitation (RIPA) buffer at pH 7.4 containing: 20 mM Tris-base, 150 mM NaCl, 1 mM EDTA, 1 mM EGTA and 1% NP40. The cell lysate was centrifuged at 14,000 rpm at 4 °C for 30 min and protein level was analyzed by Bicinchoninic acid (BCA) protein assay kit (Thermo Fisher Scientific, Waltham, MA, USA). The extraction of total membrane protein was performed by a plasma membrane protein extraction kit (BioVision, Milpitas, CA, USA). Proteins were separated on 8% or 10% sodium dodecyl sulfate polyacrylamide gel electrophoresis (SDS-PAGE) and electro-transferred onto polyvinylidene difluoride membrane. The membranes were washed with TRIS-buffered saline Tween-20 (TBST) followed by blocking with 5% skim milk for 1 h and incubated with the primary antibodies for Akt, β-actin (Santa Cruz, Santa Cruz, CA, USA); phosphorylated Akt (Ser473), GLUT4, phosphorylated insulin receptor substrate 1 (IRS1) (Ser307), IRS1, phosphorylated glycogen synthase kinase 3 (GSK3)α/β, GSK3α/β, phosphorylated p70 ribosomal protein S6 kinase (p70S6K), 70S6K, mTOR, phosphorylated mTOR (Cell Signaling, Danvers, MA, USA), p85-PI3K, phosphorylated p85-PI3K (abcam, Waltham, MA, USA), insulin receptor (IR), phosphorylated IR (Tyr960) (GeneTex, Alton Pkwy Irvine, CA, USA), and Na^+^/K^+^-ATPase (Epitomics, Burlingame, CA, USA) overnight at 4 °C. The membranes were incubated in anti-rabbit or anti-mouse antibodies conjugated with horseradish peroxidase for 1 h. The visualization of protein bands was performed by an enhanced chemiluminescence reagent (Bio-Rad, Hercules, CA, USA) and exposed to film. The densitometric analysis of blots was performed using Image J software. For immunohistochemistry, the 5-μm soleus muscle paraffin sections were used. The phosphorylated Akt in soleus muscles was stained with anti-phosphorylated Akt antibody (Cell Signaling, Danvers, MA, USA), and signals were detected by SuperPicture horseradish peroxidase polymer conjugate (Invitrogen, Carlsbad, CA, USA).

### 4.6. Transient Transfection

The transient transfection in myoblasts was performed as previously described [16]. The constitutively active form of Akt (myristoylated (myr) Akt) was a kind gift from Dr. Chen YW (China Medical University, Taichung, Taiwan). The control pcDNA3.1 empty vector and myr-Akt were transfected into myoblasts using a Lipofectamine 3000 reagent (Invitrogen, Carlsbad, CA, USA). One day following transfection, the growth medium was substituted for the differentiation medium and myoblasts were cultured for 4 days with or without acrolein. We confirmed the efficiency of transfection (about 40–50%) using an equal amount of a plasmid encoding the green fluorescent protein under the cytomegalovirus promoter.

### 4.7. Determination of Glycogen Content

The glycogen contents in myotubes were detected using a glycogen assay kit (BioVision, Milpitas, CA, USA). C2C12 myotubes were treated with 1 μM acrolein for 72 h, and then scraped in distilled water and boiled at 95℃ within 5 min. After centrifugation at 18,000 *g* for 10 min, the supernatant was collected. The glycogen contents were determined by fluorescence (Ex/Em = 535/587 nm) using a microplate fluorometer (Beckman Coulter, Brea, CA, USA).

### 4.8. Statistics

Data are presented as mean ± standard deviation. The statistical significance of differences among experimental groups was analyzed by one-way analysis of variance (ANOVA) followed by post hoc analysis using Bonferroni’s test with a significance threshold of 0.05. The SPSS statistical software (SPSS, 19.0, Chicago, IL, USA) was used.

## 5. Conclusions

Both Akt and GLUT4 are indispensable signals in myogenesis and glucose homeostasis in skeletal muscle. The findings of the present study demonstrated for the first time that acrolein at doses relevant to human exposure significantly inhibited glucose transport via the inhibition of Akt-regulated GLUT4 signaling in skeletal myotubes and impaired glucose tolerance in mice. These results suggest that acrolein may impair glucose transport and hinder glucose utilization in skeletal muscle, leading to glucose homeostatic imbalance and hyperglycemia. However, the detailed in vivo mechanism for the association between acrolein and hyperglycemia and the in vivo muscular glucose change (such as using (^18^F)FDG-PET) still needs to be clarified in the future.

## Figures and Tables

**Figure 1 ijms-22-07228-f001:**
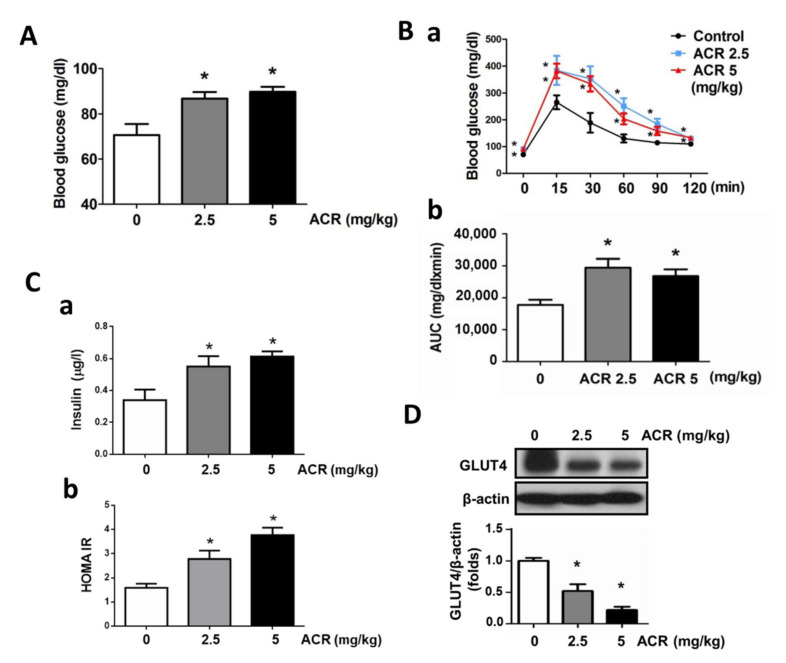
Effects of acrolein on blood glucose, glucose tolerance, blood insulin, HOMA-IR, and muscle GLUT4 protein expression in mice. Mice were treated with acrolein (2.5 or 5 mg/kg) by oral gavage daily for 4 weeks. The fasting blood glucose levels (**A**), OGTT (**B****-a**), glucose area under the curve (AUC) (**B-b**), insulin levels (**C-a**), and HOMA-IR (**C-b**) were shown. (**D**) The GLUT4 protein expression in the soleus muscles was determined by Western blotting and quantified using densitometric analysis. Results are presented as means ± SEM (n = 8). * *p* < 0.05 versus vehicle control.

**Figure 2 ijms-22-07228-f002:**
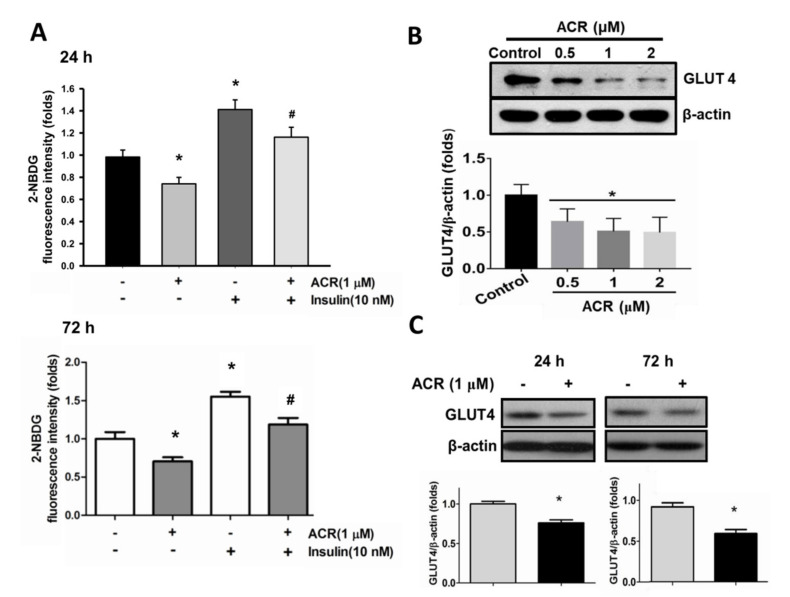
Effects of acrolein on glucose uptake and GLUT4 protein expression in differentiated C2C12 myotubes. (**A**) Myotubes were treated with 1 μM acrolein in the presence or absence of insulin (10 nM) for 24 h and 72 h. The uptake of 2-NBDG into the myotubes was evaluated by a microplate fluorometer. (**B**) The GLUT4 protein expressions in myotubes treated with various concentrations of acrolein (0.5–2 μM) for 24 h are shown. (**C**) The GLUT4 protein expressions in myotubes treated with acrolein (1 μM) for 24 h and 72 h are shown. The protein expression was determined by Western blotting and quantified using densitometric analysis. Results are represented as means ± SEM for at least four independent experiments. * *p* < 0.05 versus vehicle control; # *p* < 0.05 versus acrolein alone.

**Figure 3 ijms-22-07228-f003:**
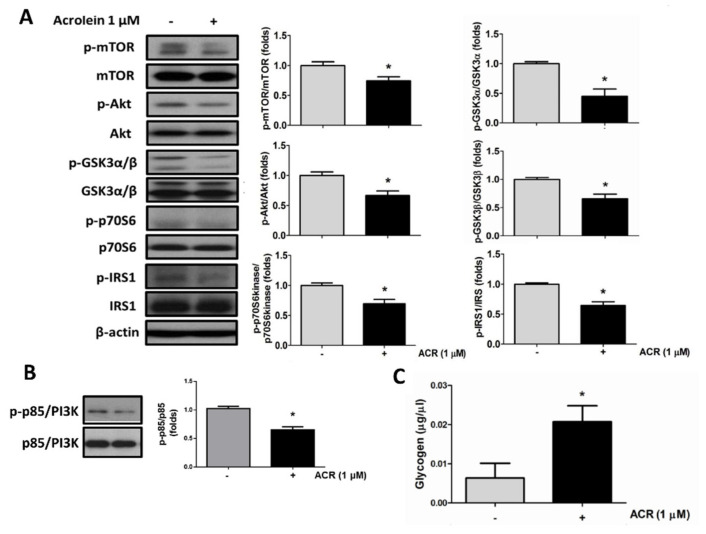
Acrolein interferes with glucose metabolic signaling molecules in differentiated C2C12 myotubes. Myotubes were treated with 1 μM acrolein for 24 h. (**A**) The levels of phosphorylated and total protein expression of IRS1, Akt, mTOR, p70S6K, and GSK3α/β were determined by Western blotting and quantified using densitometric analysis. (**B**) The levels of phosphorylated and total protein expression of p85/PI3K were determined by Western blotting and quantified using densitometric analysis. (**C**) The glycogen contents in myotubes treated with acrolein (1 μM) for 72 h are shown. Results are represented as means ± SEM for at least four independent experiments. * *p* < 0.05 versus vehicle control.

**Figure 4 ijms-22-07228-f004:**
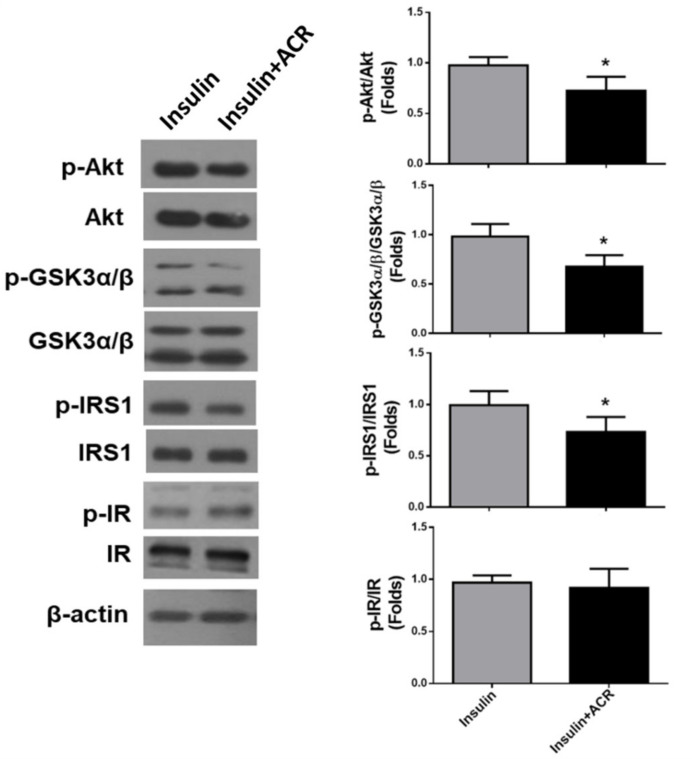
Acrolein interferes with insulin signaling in differentiated C2C12 myotubes. Myotubes were treated with 1 μM acrolein for 24 h and then stimulated with insulin (10 nM) for 10 min. The levels of phosphorylated and total protein expression of Akt, GSK3α/β, IRS1, and IR were determined by Western blotting and quantified using densitometric analysis. Results are represented as means ± SEM for at least four independent experiments. * *p* < 0.05 versus insulin alone.

**Figure 5 ijms-22-07228-f005:**
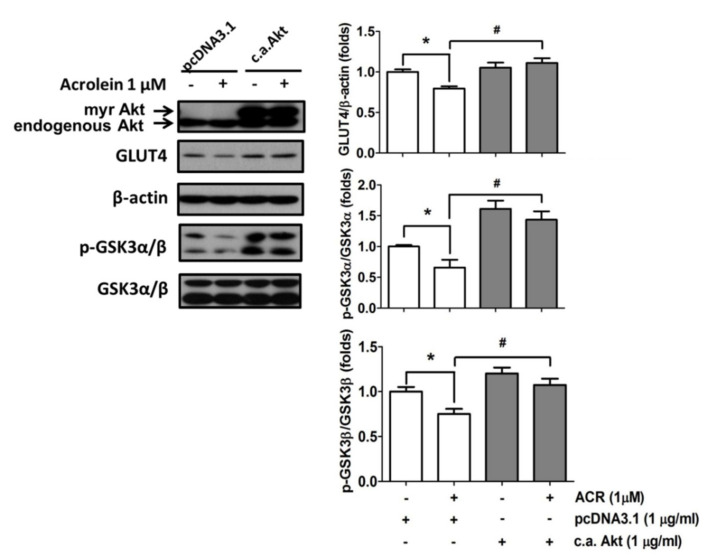
Over-expression of the constitutively active form of Akt ameliorates the inhibitory effects of acrolein on the protein expressions of GLUT4 and phosphorylated GSK3α/β in myotubes. Myotubes were transiently transfected with control pcDNA3.1 or c.a. Akt for 24 h, and then exposed to 1 μM acrolein for 24 h. The protein expressions of total GLUT4 and phosphorylated GSK3α/β were determined by Western blotting and quantified using densitometric analysis. Results are represented as means ± SEM for at least four independent experiments. * *p* < 0.05 versus vehicle control; # *p* < 0.05 versus acrolein + pcDNA3.1.

**Figure 6 ijms-22-07228-f006:**
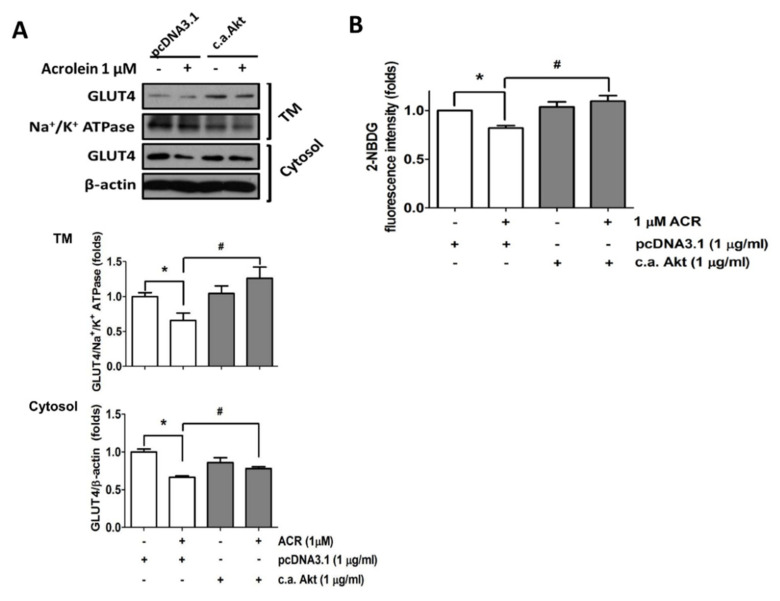
Over-expression of the constitutively active form of Akt ameliorates the inhibitory effects of acrolein on GLUT4 and glucose uptake in myotubes. Myotubes were transiently transfected with control pcDNA3.1 or c.a. Akt for 24 h, and then exposed to 1 μM acrolein for 24 h. (**A**) The protein expressions of GLUT4 from total membrane (TM) fraction and cytosol fraction were determined by Western blotting and quantified using densitometric analysis. (**B**) The uptake of 2-NBDG into the myotubes was evaluated by a microplate fluorometer. Results are represented as means ± SEM for at least four independent experiments. * *p* < 0.05 versus vehicle control; # *p* < 0.05 versus acrolein + pcDNA3.1.

**Figure 7 ijms-22-07228-f007:**
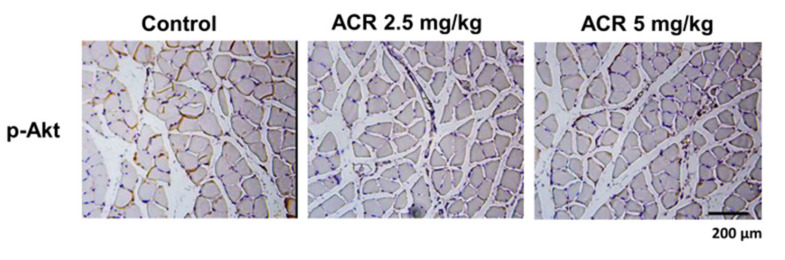
Effects of acrolein on the expression of phosphorylated Akt in the soleus muscles of mice. Mice were treated with acrolein (2.5 or 5 mg/kg) by oral gavage daily for 4 weeks. The representative immunohistochemical images for phosphorylated Akt expression in the soleus muscles isolated from each group are shown.

## Data Availability

The data presented in this study are available from the corresponding author upon reasonable request.

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
