# Peer review of "Low-Dose Acrolein, an Endogenous and Exogenous Toxic Molecule, Inhibits Glucose Transport via an Inhibition of Akt-Regulated GLUT4 Signaling in Skeletal Muscle Cells"

_ijms, 2021, doi:10.3390/ijms22137228_

Round 1

Reviewer 1 Report

In this manuscript, Wang et al reported the negative effects of acrolein at doses relevant to human exposure in glucose tolerance, levels of GLUT4 (both in vivo and in vitro), glucose uptake in C2C12 myotubes, as well as in the phosphorylation of relevant proteins of the insulin signaling cascade. A step further, they demonstrated that constitutive activation of Akt counteracts these molecular alterations.

General comment

Although of potential interest in the field on insulin resistance induced by environmental chemicals, in its present form this manuscript is preliminary and also contains some inconsistences.

Specific comments

-The authors demonstrated decreased GLUT4 in skeletal muscle of mice treated with acrolein, as well as in C2C12 myotubes exposed to this chemical. Since skeletal muscle glucose uptake largely contributes to glucose clearance, insulin tolerance test (ITT) must be performed in the mouse model.

-In Figure 2A it is not clear whether acrolein was added to C2C12 myotubes in the absence or presence of insulin. At 24 h it seems that it is added alone, but at 72 h the figure indicates that acrolein is added together with insulin. Nevertheless, the authors need to demonstrate the impact of acrolein in insulin-induced glucose uptake by treating the cells with this chemical for 23-72 h and then stimulate with insulin and evaluate glucose uptake.

-In line with the previous comment, the effect of acrolein in insulin signaling at least in C2C12 myotubes must be included in this study. It is important to include the phosphorylation of the insulin receptor

-An important inconsistency in Figure 3A is the evaluation of IRS1 Ser 307 phosphorylation (this is the antibody found in Materials and Methods, line 273). It has been extensively shown that IRS1 Ser 307 phosphorylation impairs insulin signaling by decreasing IRS1 tyrosine phosphorylation. Therefore, if acrolein decreases IRS1 Ser 307 phosphorylation (Figure 3A) the rationale is to increase insulin signaling. This must be clarified.

-The expression and phosphorylation of these relevant insulin signaling mediators must be addressed in skeletal muscle from mice receiving acrolein treatment

-The explanation for the increased glycogen content by acrolein is not convincing.

Author Response

Response to the comments by Reviewer 1:

In this manuscript, Wang et al reported the negative effects of acrolein at doses relevant to human exposure in glucose tolerance, levels of GLUT4 (both in vivo and in vitro), glucose uptake in C2C12 myotubes, as well as in the phosphorylation of relevant proteins of the insulin signaling cascade. A step further, they demonstrated that constitutive activation of Akt counteracts these molecular alterations.

General comment

Although of potential interest in the field on insulin resistance induced by environmental chemicals, in its present form this manuscript is preliminary and also contains some inconsistences.

Response: We appreciate the reviewer's comment. We have done our best to improve these issues according to the suggestions of reviewer. This study found the inhibition of GLUT4 protein expression in muscles of acrolein-treated mice and tried to investigate its possible mechanism using an in vitro myotube model. We found that the inhibition of Akt-regulated GLUT4 signaling may play an important role in acrolein-inhibited glucose uptake. To strength our findings, we added the data for insulin levels and HOMA-IR to support that acrolein could induce insulin resistance in mice. We also added the data for the expression of phosphorylated Akt in soleus muscle of acrolein-treated mice to support that acrolein could effectively decrease the Akt phosphorylation in vitro and in vivo. We also added the data for the expression of phosphorylated p85/PI3K in acrolein-treated myotubes to confirm the effect of acrolein on PI3K/Akt pathway. We also explained and discussed the issues for IRS1 ser307 phosphorylation, glycogen content, and others in this revised manuscript according to the suggestions of reviewer.

Specific comments

-The authors demonstrated decreased GLUT4 in skeletal muscle of mice treated with acrolein, as well as in C2C12 myotubes exposed to this chemical. Since skeletal muscle glucose uptake largely contributes to glucose clearance, insulin tolerance test (ITT) must be performed in the mouse model.

Response: We appreciate the reviewer's comment. We sincerely hope that the reviewer can understand that we are currently unable to perform a new animal experiment, because Taiwan is currently in COVID-19 outbreak and there is the level 3 epidemic alert for Taipei City. Nevertheless, we used the reserved blood samples for analysis of insulin and HOMA-IR. We added the data for blood insulin levels and HOMA-IR in this revised manuscript. Acrolein (2.5 and 5 mg/kg) exposure could significantly increase the insulin levels and HOMA-IR (Figure 1C of this revised manuscript).

-In Figure 2A it is not clear whether acrolein was added to C2C12 myotubes in the absence or presence of insulin. At 24 h it seems that it is added alone, but at 72 h the figure indicates that acrolein is added together with insulin. Nevertheless, the authors need to demonstrate the impact of acrolein in insulin-induced glucose uptake by treating the cells with this chemical for 23-72 h and then stimulate with insulin and evaluate glucose uptake.

Response: We appreciate the reviewer's comment. We repeated the experiment of glucose uptake at 24 h according to the suggestion of reviewer. Insulin (10 nM) was as a positive control. The glucose uptake determined by 2-NBDG assay was significantly decreased in myotubes treated with acrolein for 24 and 72 h (Figure 2A of this revised manuscript). The results from 24 h treatment are similar to the results from 72 h treatment.

-In line with the previous comment, the effect of acrolein in insulin signaling at least in C2C12 myotubes must be included in this study. It is important to include the phosphorylation of the insulin receptor

Response: We appreciate the reviewer's comment. As the comment of the reviewer for effect of acrolein on IRS1 Ser 307 phosphorylation, there may be a controversial effect for acrolein on insulin receptor-IRS1 signaling. The detailed effects and mechanisms of acrolein on the inhibition of IRS1 Ser(P) 307 and insulin receptor function need further investigation in the future. The present study exactly focused on the role of PI3K/Akt in acrolein-inhibited GLUT4-related glucose uptake. We, therefore, used an in vitro myotube model to investigate the molecular mechanism of acrolein for this aim. Nevertheless, we added the data for the expression of phosphorylated p85/PI3K in acrolein-treated myotubes to confirm the effect of acrolein on PI3K/Akt pathway. The phosphorylation of p85/PI3K in myotubes could also be reduced by acrolein (1 μM) treatment for 24 h (Figure 3B of this revised manuscript).

-An important inconsistency in Figure 3A is the evaluation of IRS1 Ser 307 phosphorylation (this is the antibody found in Materials and Methods, line 273). It has been extensively shown that IRS1 Ser 307 phosphorylation impairs insulin signaling by decreasing IRS1 tyrosine phosphorylation. Therefore, if acrolein decreases IRS1 Ser 307 phosphorylation (Figure 3A) the rationale is to increase insulin signaling. This must be clarified.

Response: We appreciate the reviewer's comment. We tried to clarify and discuss this issue in this revised manuscript according to the suggestion of reviewer.

This is a controversial issue. We discussed this issue in the Discussion of this revised manuscript as follows:

Phosphorylation of IRS1 Ser307 has been shown to inhibit the interaction between IRS1 and insulin receptor, leading to inhibit insulin action in a 32D myeloid progenitor cell model [30]. Hançer et al. have found that three of the IRS1 Ser(P) residues 302, 307, and 612 are negatively correlated with tyrosine phosphorylation of IRS1 in a Chinese hamster ovary-insulin receptor/IRS1 overexpression cell model, showing an insulin-desensitizing effect [31]. However, contrary to the results from experiments of cell models, Copps et al. found that IRS1 Ser307 played a positive regulatory role in maintaining insulin signaling in a separate knock-in mouse model, resulting in moderating the severity of insulin resistance [32]. Therefore, the more evidence is needed to clarify this controversial issue. On the other hand, Hançer et al. have shown that the inhibition of the PI3K-Akt-mTOR pathway by pharmacological inhibitors can consistently inhibit IRS1 Ser(P) 265, 302, 307, 522, and 632 [31]. The present study focused on the role of PI3K/Akt in acrolein-inhibited GLUT4-related glucose uptake. Our results revealed that acrolein inhibited the activation of PI3K-Akt-mTOR signaling pathway and the phosphorylation of IRS1 Ser307 in myotubes. These findings emphasized the important role of PI3K/Akt signaling in acrolein-impaired glucose uptake in skeletal muscles. However, the detailed effects and mechanisms of acrolein on the inhibition of IRS1 Ser(P) 307 and insulin receptor function need further investigation in the future.

-The expression and phosphorylation of these relevant insulin signaling mediators must be addressed in skeletal muscle from mice receiving acrolein treatment

Response: We appreciate the reviewer's comment. Because these relevant insulin signaling mediators were not easy to be addressed in skeletal muscles from low-dose acrolein-treated mice, we, therefore, used an in vitro myotube model to investigate the possible molecular mechanism of acrolein on muscles. We found that the inhibition of Akt-regulated GLUT4 signaling played an important role in acrolein-induced glucose uptake damage. Nevertheless, we tried to clarify whether the Akt phosphorylation was inhibited by acrolein in vivo. As shown in Figure 6 of this revised manuscript, the immunohistochemistry staining showed that the soleus muscle isolated from acrolein-treated mice had a decreased phosphorylated Akt protein expression.

-The explanation for the increased glycogen content by acrolein is not convincing.

Response: We appreciate the reviewer's comment. We tried to explain and discuss this issue in this revised manuscript according to the suggestion of reviewer.

This is a controversial issue. We discussed this issue in the Discussion of this revised manuscript as follows:

The present study found that low-concentration acrolein significantly inhibited the phosphorylation of GSK3α/β in C2C12 myotubes, which could be effectively reversed by transfection of the constitutively active form of Akt. These results suggest that the inhibition of Akt-regulated GSK3α/β phosphorylation by acrolein activates GSK3 activity, which may further retard glucose transport and glycogen synthesis in myotubes. However, we observed a controversial increase in glycogen content in acrolein-treated myotubes. Therefore, the increased glycogen content by acrolein in myotubes cannot be explained by increase in the activity of classical insulin-signaling pathway (PI3K-Akt-GSK3). These results indicate that other pathways may be involved in the modulation of glycogen synthase in acrolein-treated myotubes. The glycogen content in muscles has been found to be effectively increased in GLUT4 null mice [37]. Kim et al. have also found that the muscular glycogen synthase activity and glycogen content in muscle-specific GLUT4 knockout (KO) mice are markedly elevated despite a 75% reduction in glucose uptake [38]. They found that PI3K and Akt activities were decreased and GSK3β activity was increased, predicting that glycogen synthesis would be inhibited in GLUT4-KO muscles; however, the glycogen content was exactly increased [38]. They further demonstrated that the increased hexokinase II, glucose-6-phosphate, muscle-specific regulatory subunit (RGL), protein targeting to glycogen (PTG) levels and enhanced glycogen-targeting subunits of protein phosphatase 1 (PP1) activity might contribute to the increased glycogen synthase activity and glycogen content [38]. Acrolein has been shown to increase the liver glycogen content of fasted rats that may be associated with the increased adrenal glucocorticoids, leading to stimulate gluconeogenesis and glycogenesis [39]. Our findings revealed that acrolein exposure significantly decreased GLUT4 protein expression and glucose uptake, but markedly increased glycogen contents in myotubes, suggesting that low-concentration acrolein exposure may retard glucose metabolism and evoke the compensative glycogen synthesis in myotubes, which may be independent of the PI3K-Akt-GSK3 pathway. Nevertheless, the real mechanism for this issue needs to be clarified in the future.

Reviewer 2 Report

The purpose of this study is to discover the effect of acrolein on muscular glucose metabolism and glucose tolerance in skeletal muscle in vitro and in vivo, respectively. To demonstrate this, the authors investigated the fasting blood glucose levels, oral glucose tolerance test, and measurements of GLUT4 expression in the skeletal muscle after exposure of mice to acrolein, including transient transfection and glycogen content experiments. The introduction is well written and provides sufficient information with adequate references, the methodology and experimental design are appropriate, results are clearly presented, and conclusion are supported by the results. However, it seems that minor English correction is necessary through the whole manuscript to provide more helpful information. Taken as a whole, this manuscript could be accepted for publication in International Journal of Molecular Sciences after the following suggested minor changes are made:

  1. I suggest that several sentences starting with “we~” be modified throughout the manuscript, especially result section.
  2. in line 211, remove “(2005)”
  3. in line 230, more detailed information, such as sampling method used (tail tip?), anesthesia condition and fasting time, should be provided the blood sampling process. It is known that some differences can occur depending on the blood sampling condition.
  4. in line 242, 2 in CO2 is a subscript
  5. in line 256, were measured
  6. in line 270, were washed
  7. in line 301, I think it should be described as mean +/- standard deviation, not SEM
  8. in line 321-323; Careful typo checking is required.

In addition, I think the identification of glucose changes using [18F]FDG-PET under in vivo conditions could yield more quantitative and non-invasive information can be obtained, which is thought to provide solid support for the conclusion. 

Author Response

Response to the comments by Reviewer 2:

The purpose of this study is to discover the effect of acrolein on muscular glucose metabolism and glucose tolerance in skeletal muscle in vitro and in vivo, respectively. To demonstrate this, the authors investigated the fasting blood glucose levels, oral glucose tolerance test, and measurements of GLUT4 expression in the skeletal muscle after exposure of mice to acrolein, including transient transfection and glycogen content experiments. The introduction is well written and provides sufficient information with adequate references, the methodology and experimental design are appropriate, results are clearly presented, and conclusion are supported by the results. However, it seems that minor English correction is necessary through the whole manuscript to provide more helpful information. Taken as a whole, this manuscript could be accepted for publication in International Journal of Molecular Sciences after the following suggested minor changes are made:

  1. I suggest that several sentences starting with “we~” be modified throughout the manuscript, especially result section.
  2. in line 211, remove “(2005)”
  3. in line 230, more detailed information, such as sampling method used (tail tip?), anesthesia condition and fasting time, should be provided the blood sampling process. It is known that some differences can occur depending on the blood sampling condition.
  4. in line 242, 2 in CO2 is a subscript
  5. in line 256, were measured
  6. in line 270, were washed
  7. in line 301, I think it should be described as mean +/- standard deviation, not SEM 8. in line 321-323; Careful typo checking is required. 
  8. In addition, I think the identification of glucose changes using [18F]FDG-PET under in vivo conditions could yield more quantitative and non-invasive information can be obtained, which is thought to provide solid support for the conclusion. 

Response: We appreciate the reviewer's comment. We have revised the manuscript according to the suggestions of reviewer. We also agree with the comment suggested by reviewer for identification of glucose changes using [18F]FDG-PET under in vivo conditions; we have added this comment in the Conclusion section for one of the limitations of our study.

Round 2

Reviewer 1 Report

The authors have addressed some, but not all, my comments. Among all poits that I raised in my previous revision a key important issue is still missing

The effect of acrolein in insulin signaling at least in C2C12 myotubes must be included in this study. Cells must be cultured for 24-72 h with actolein and then stimulated with insulin for 10 min and the key responses of insulin signaling (phosphorylation of IR, IRS1, Akt, GSK3) need to be evaluated by Western blot

Author Response

Reviewer 1: (round 2)

The authors have addressed some, but not all, my comments. Among all poits that I raised in my previous revision a key important issue is still missing

The effect of acrolein in insulin signaling at least in C2C12 myotubes must be included in this study. Cells must be cultured for 24-72 h with actolein and then stimulated with insulin for 10 min and the key responses of insulin signaling (phosphorylation of IR, IRS1, Akt, GSK3) need to be evaluated by Western blot

Response: We appreciate the reviewer's comment. We have added the data for the effect of acrolein in insulin signaling in C2C12 myotubes (revised Fig. 4) in this revised manuscript according to the suggestion of reviewer.

We examined the effects of acrolein in insulin signaling in C2C12 myotubes. We have shown that acrolein treatment for 24 h can significantly decrease glucose uptake in myotubes with or without insulin stimulation (Fig. 2) and inhibit the glucose uptake-related signaling molecules expression in myotubes (Fig. 3). We, therefore, selected a 24-hour duration for this additional experiment. Myotubes were cultured for 24 h with acrolein (1 μM), and then stimulated with insulin (10 nM) for 10 min. As shown in Figure 4, the levels of phosphorylation for IRS1, Akt, and GSK3α/β, but not insulin receptor (IR), were significantly reduced by acrolein (1 μM) treatment in insulin-stimulated myotubes.

These findings indicated that acrolein exposure could interfere with the insulin signaling in myotubes. However, the detailed effects and mechanisms of acrolein on the inhibition of IRS1 Ser(P)307 and insulin receptor function need further investigation in the future.

Round 3

Reviewer 1 Report

The authors have addressed all my comments